# Protocol for an observational study of working conditions and musculoskeletal health in Swedish online retail warehousing from the perspective of sex/gender and place of birth

**Jennie A. Jackson**[1]*, **Svend Erik Mathiassen**[1], **Klara Rydström**[2], **Kristina Johansson**[2]

**1** Centre for Musculoskeletal Research, Department of Occupational Health Science and Psychology, University of Gävle, Gävle, Sweden, **2** Department of Social Sciences, Technology and Arts, Luleå University of Technology, Luleå, Sweden

* jennie.jackson@hig.se

**Data Availability Statement:** No datasets were generated or analysed during the current study. All

## Abstract

European and International sustainable development agendas aim to reduce inequalities in working conditions and work-related health, yet disparate occupational health outcomes are evident between both men and women and domestic- and foreign-born workers. In Sweden, major growth in online retail warehousing has increased occupational opportunities for foreign-born workers. The rapid change has left research lagging on working conditions, i.e., employment conditions, facility design, work organisation, physical and psychosocial work environment conditions, and their effects on worker health. Further, no known studies have considered patterns of inequality related to these factors. The overall aim of this study is to describe working conditions and musculoskeletal health in online retail warehousing, determine the extent to which differences exist related to sex/gender and place of birth (as a proxy for race/ethnicity), and examine factors at the organisational and individual levels to understand why any differences exist. Three online retail warehouses, each employing 50–150 operations workers performing receiving, order picking, order packing and dispatching tasks will be recruited. Warehouses will, to the extent possible, differ in their extent of digital technology use. Employment conditions, facility design (including digital tool use), work organisation, physical and psychosocial work environment conditions and worker health will be assessed by survey, interview and technical measurements. Analysis of quantitative data stratified by sex and place of birth will consider the extent to which inequalities exist. Focus group interviews with operations employees and in-depth interviews with managers, union and health and safety representatives will be conducted to assess how employee working conditions and musculoskeletal health are related to inequality regimes of sex/gender and/or race/ethnicity in organisational processes and practices in online retail warehousing. The study is pre-registered with the Open Science Framework. This study will describe working conditions and health in online retail warehouse workers and consider the extent to which patterns of inequality exist based on sex/gender and place of birth.

relevant data from this study will be made available upon study completion.

**Funding:** KJ & SEM - Swedish Research Council for Health, Working Life and Welfare (Forte) grant number 2019-01051. https://forte.se/ The funders did not and will not have a role in study design, data collection and analysis, decision to publish, or preparation of the manuscript.

**Competing interests:** The authors have declared that no competing interests exist.

## Introduction

Reducing inequalities in working conditions and work-related health based on sex/gender, race, ethnicity and (im)migration status is necessary to meet the goals laid out in current sustainable development agendas globally [1], within the European Union [2], and within the Swedish National Work Environment Strategy [3]. The goals reflect current and past trends in work-related health disparities. Over the last thirty years, differences have been reported in occupational health between males and females (i.e. sex differences between groups based on physiological characteristics), for example, the annual incidence of musculoskeletal disorders (MSDs) in Europe has been approximately thirty percent higher for females than males [4], with particularly pronounced differences in the upper extremity MSD rates [5–7]. In contrast, the relationships between occupational health and gender—that is, a person's socioculturally constructed identity as man, woman, or non-binary—or race/ethnicity—that is, social constructs related to physical traits (i.e. race) and sociocultural characteristics (i.e. ethnicity) [8]—have not yet been systematically investigated across Europe or within Sweden. Swedish data suggest foreign-born (FB) workers have a lower work-related health status than Swedish-born (SB) workers [9] including higher rates of poor self-reported health and mental distress [10], and higher rates of sick leave absence [9, 11] and burnout [12]. The imbalances in work-related health related to place of birth are particularly concerning in the Nordic context given the steep increase in immigration rates in the last decade [13]. Research further shows that the labour market is segregated and segmented based on both sex/gender and race/ethnicity [14], and may include a more complex hierarchy of inequality based on intersections between sex/gender and race/ethnicity [15]. The expansion of the consideration of equality from strictly male-female (sex) differences to also include consideration of gender, and, further, of race, ethnicity and (im)migration reflects the reality of modern culture and immigration patterns and underscores the need for intersectionality, that is, the simultaneous consideration of different aspects of identity in understanding inequality in working conditions and health outcomes [16]. Labour market segregation is well documented in Europe, including Sweden [17, 18], and the resulting differential occupational exposures and risk factors between occupations are likely to contribute highly to disparate national health outcomes related to sex/gender and race/ethnicity. However, health disparities are still evident across workers within same occupational sectors, with similar evidence of inequalities based on both sex/gender [19–21] and race/ethnicity [22].

Warehouses play a key role in the global economy. Given the current competitive markets, companies must be able to deliver products quickly and with high-quality service with minimal costs making warehousing a key element in overall company business processes. To increase the performance of warehouse operations systems, digital technology use in warehousing is increasing. For example, mobile devices for the augmentation of processes and support of workers in order picking are now commonplace, and the use of collaborative robots, for example where parts are delivered to the workers who pack the orders (for example, Ocado and Modula vertical warehouse solutions [23, 24]), is on the rise. Changes in digital technology will lead to changes in facility design and, likely, in work environment conditions and worker health. Warehousing in online retail, defined as 'retail sale via mail order houses or via Internet' [25], in which orders are picked and packed for customers and, in some cases, for physical stores, has experienced rapid growth both in terms of numbers of employees and adoption of digital technologies. In Sweden, between 2015–2020, over 7000 new employees were hired into online retail, which represented a 3% increase in the share of total employees in retail trade (from 4% to 7%) [25, 26]. Changes to the way human work is organized and performed in warehousing are evident [27]. In a recent report, the Swedish Work Environment Authority

described online retail operations systems as having been designed to create an optimal flow of goods to the customer, but with little consideration of the working environment for the employees [28]. The rapid growth of online retail has left work environment research lagging regarding assessment of the working conditions and health in online retail warehousing.

Unlike traditional retail work environments, warehouse work typically requires no interaction with customers. In Sweden, this can translate to reduced language proficiency requirements in warehousing compared to traditional retail jobs and subsequently increased accessibility for foreign-born workers. Further, the 'behind-the-scenes' nature of warehousing likely removes the traditional 'light skinned, young, middle-class, heterosexual femininity' architype that is preferentially employed in front-line service jobs [29–31]. Thus, an increase in both foreign-born and male employees would be expected in online retailing compared to traditional retail jobs. Labor statistics based on administrative sources from Statistics Sweden confirm this prediction. In 2020, 24% of online retail workers were foreign-born compared with 17% in traditional retail, and 50% of online retail workers were men compared with 38% in traditional retail. The diverse worker population in online retail warehousing is likely to face embedded practices of inequalities related to gender and/or race/ethnicity, as described in Acker's (2006) inequality regimes, which can lead to differences in work environment conditions and health. In our recent review of work conditions in warehousing in relation to gender and race/ethnicity [15], we found evidence that gender and race/ethnicity influenced work organisation, work environment conditions, and employment conditions. The results also underline the need for research considering patterns of gender and race/ethnicity in working conditions and health in online retail warehousing.

Disparate health outcomes related to sex/gender and race/ethnicity are likely a result of a combination of societal, organisational, and individual level factors. A framework for a hierarchical set of factors underlying sex/gender inequalities and musculoskeletal health was previously proposed which included the effects of differences due to: occupation, work tasks within occupation, loads within work tasks, load effects, reporting behaviours, and treatment attitudes [6, 32]. The present study expands that theoretical framework to include both physical and psychosocial work environment conditions in relation to both sex/gender and race/ethnicity and focuses specifically on the factors, work tasks, and, loads in tasks, in the assessment of inequalities in online retail warehousing. Differences in allotment of tasks according to gender and/or race/ethnicity, as part of the work organisation among employees with the same job title, can result in differences in tasks performed by groups of workers and subsequently differences in occupational exposures and risks. For example, research shows that women typically perform tasks that are repetitive and less variable than those performed by men [33, 34], and that FB workers have more physically demanding work and more awkward working postures [10] and are more likely to perform tasks with a higher risk than domestic-born workers [9]. Even among workers performing the same tasks, the loads experienced may differ between workers. For example, the physical loads experienced by an average sized woman will likely differ compared with an average size man when performing work at a fixed workstation or with a specific tool. Loads experienced could be higher for women given workstations and tools are typically designed based on the anthropometrics and capacity of an average male [35, 36]. Differential experiences of psychosocial loads have also been documented, for example, between FB and domestic-born workers where FB workers reported lower workplace decision latitude [10, 37] and social support [10, 38].

According to Acker's theory of inequality regimes, 'loosely interrelated practices, processes, actions and meanings' exist in all organisations and create hierarchies of gendered and racialized class relations [39]. Acker's theory considers that relationships and power balances between members in an organisation are therefore not only hierarchically ordered according

to occupational rank, e.g., CEO, manager, middle management, production workers, but are also gendered and racialized. We use the term 'inequality' to refer to "systematic disparities between participants" [39] that are associated with employee sex/gender and race/ethnicity. Systematic disparities can occur at all levels of organisational processes and practices, for example, in preferential recruitment that targets a certain category (e.g. sex, gender, race, ethnicity) of worker deemed most suitable for a task, in systematic promotion and wage disparity practices, and/or in informal interactions between employers and employees. We used the term race/ethnicity to encompass aspects related to where a person is from and their sociocultural identity, including pre-conceived notions a person encounters regarding their origin and abilities as related to their physical appearance. In the quantitative analyses, we use place of birth (PoB) as a proxy for race/ethnicity. While PoB and sociocultural identity typically interact, we acknowledge that race, ethnicity, place of birth, (im)migration status and language development are differentiated both in theory and lived experiences (cf. [8, 40]). Specifically, we accept that "race is a socially constructed bodily concept, whereas ethnicity is a socially constructed cultural concept" [8], and that country of birth and (im)migration status are directly measurable quantities. In this study, comparisons related to PoB and race/ethnicity will specifically reflect differences in experiences between Swedish- and foreign-born workers.

The overall aim of this 3-year study is to describe working conditions and musculoskeletal health in online retail warehousing including the extent to which differences based on sex/gender and race/ethnicity exist in working conditions and health, and to examine factors at the organisational and individual levels underlying eventual inequalities in working conditions and health. Fig 1 shows the conceptual model underlying the current study, and illustrates the link between working conditions and health and the potential for differences in working conditions and health based on sex/gender and race/ethnicity. In the model, examples are given of specific factors that will be investigated in the present study. We use the term 'working conditions' as an overarching term for the factors represented on the left side of the model (Fig 1), which includes the employment conditions of the warehousing workers as well as the facility design, work organisation and work environment conditions of the warehouse operations system. Employment conditions refers to the type of employment (regular versus temporary; full-time versus part-time), the extent of work (full-time or part-time), wage levels and benefits.

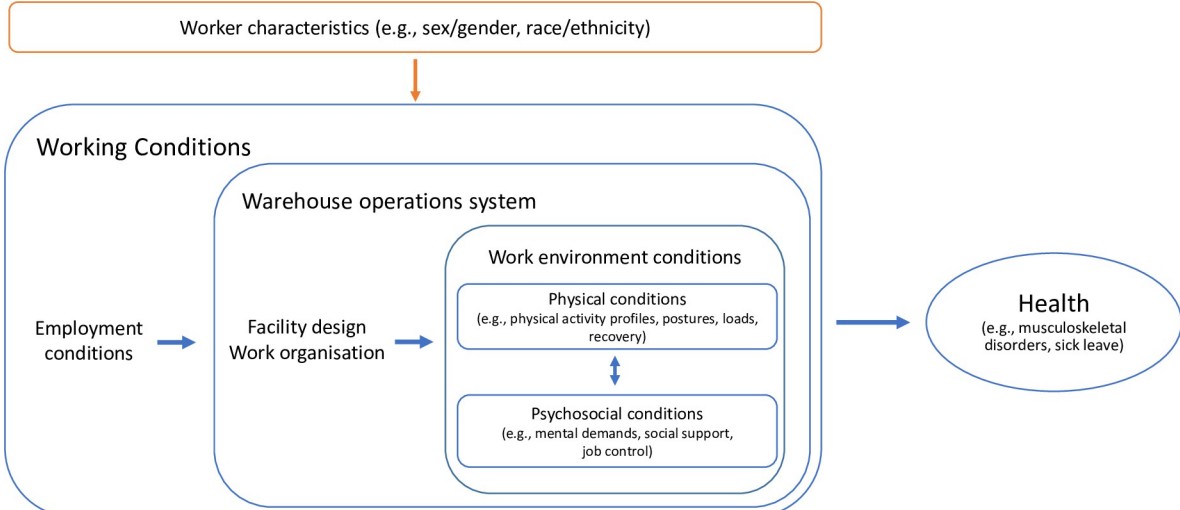

**Fig 1. Working conditions and health in relation to sex/gender and race/ethnicity.** Conceptual model outlining the linkages investigated in the current study between working conditions and health including consideration of individual worker characteristics.

Facility design includes the physical components of the operations system underlying the operations process flow including digital production technologies. Work organisation considers how work is organized between employees, including the allotment of tasks. Facility design and work organisation both influence the physical and psychosocial work environment conditions, which interact with one another.

The present study aims to both describe the extent to which differences in working conditions and health exist based on sex/gender and PoB (as a proxy for race/ethnicity) as well as to provide an understanding of why any differences exist by assessing inequality regimes. To achieve this goal, the study will integrate results from quantitative analyses of work environment conditions and health and qualitative analyses of the organisation's inequality patterns in employment conditions and work organisation to (i) identify and describe factors within the organisation that affect working conditions, and (ii) understand how the organisation creates and maintains these conditions. A mixed methods approach involving both quantitative and qualitative tools is therefore required. By both identifying patterns of inequality and assessing how the patterns occur, the results will provide evidence regarding the possibilities and obstacles for increasing equality in working conditions and health.

The specific study aims are to:

1. Describe working conditions and work-related musculoskeletal health in online retail warehousing.

2. Determine the extent to which working conditions and work-related musculoskeletal health differ between employees with respect to sex/gender and PoB in online retail warehousing.

3. Describe the extent to which sex/gender and/or PoB inequalities in working conditions and musculoskeletal health are related to organisational processes and practices in online retail warehousing.

## Materials and methods

Inequalities in working conditions and health will be assessed on the basis of sex/gender and PoB at three online retail warehouses located in Sweden using a mixed-methods approach including questionnaires, technical measurement, focus-group interviews with production employees, and semi-structured interviews with managers, and union and health and safety representatives.

### Selection of warehouses

In preparing for this study, we conducted site visits to warehouses to improve our understanding of operations systems including facility design, use of digital technology, workflow, types of goods handled, and workforce size and demographics, including the relative proportions of male and female and Swedish- and foreign-born operations employees. These visits guided our selection criteria and recruitment of warehouses.

In Sweden, approximately one third of online retail warehouse workers are employed in warehouses with less than 20 employees, one third in warehouses with 20–200 employees, and one third in warehouses with more than 200 employees (Statistics Sweden, 2020). We will focus on warehouses with 20–200 employees, aiming specifically for 50–150 operations employees.

In general, the warehousing goods handling process can be divided into receiving (including unpacking, registering, and sorting, and allocation of goods to shelves); order picking; order packing; and dispatching of the outgoing goods (including palletizing of packed orders

and transfer of pallets to loading areas). The size and mass of goods handled will directly impact the loads experienced by workers. We will focus on warehouses dealing predominantly with smaller sized pick item items under 5 kg but will also include and measure on workers assigned to the unique warehouse areas where heavy or large item packing is performed.

The extent of automation varies widely between warehouses. We aim to recruit warehouses spanning the range of digital technologies. We will describe the extent of digital technology usage at each warehouse qualitatively. Our analyses will include consideration of possible associations between digital technology use and inequality based on gender and race/ethnicity.

## Study population

All permanent and contract production employees will be invited to participate in all three data collection methods included in the study (survey, technical measurement, and focus group interview). We aim to collect survey data assessing aspects of both the physical and psychosocial work environments from at least 70% of production employees (approx. 300 in total), technical measurements of working postures and heart rate variability from 60 employees (20 per site) and conduct focus group interviews with at least 20 workers per warehouse. In addition, in-depth individual interviews with union and health and safety representatives and production and site managers will be conducted at each site.

## Recruitment

Recruitment of warehouses will occur during autumn of 2022 and be done via email followed by digital meetings between researchers and company representatives including human resources, management, union and health and safety delegates.

For recruitment of production participants at each warehouse, we aim to reach employees directly via an in-person presentation of our study. To this end, a minimum 1-day recruitment trip will be made by researchers to each warehouse in advance of the data collection. During the recruitment day(s) researchers will meet with human resources, union and health and safety representatives, and production and site managers to discuss practical details of the study and, to the extent possible, inform production employees via group presentations and/or by being present in worker coffee break rooms, equipped with posters and brochures to garner interest. All employees with whom we have contact will be asked to complete an 'intent to participate' form indicating the part(s) of the study in which they would like to participate, which work tasks they typically perform, and their working hours during the data collection week. For the technical measurements and focus group interviews, participants will be purposively selected from the pool of interested parties working day or afternoon shifts to create the best possible balance between males and females, Swedish-born (SB) and foreign-born (FB) workers, while also including, to the extent possible, a wide a range of worker competences, lengths of employment, ages and areas/tasks performed.

## Data collection

All production employee data collection will be performed during a one-week period. Technical measurements will run Monday to Friday; employees will be offered a chance to complete the questionnaire (online or paper format) between Tuesday and Friday, and focus group interviews will be conducted in the later days of the week.

To the extent possible, individual in-depth interviews with managers, health and safety and union representatives will be conducted in-person during the same week as production employee data are collected. Additional interviews will be conducted online as necessary.

Data collection will occur during autumn 2022 and winter 2023. The study received approval from the Swedish Ethical Review Authority (Etikprövningsmyndigheten)—reference number 2020–03237. Written informed consent will be obtained from all participants in the questionnaire and technical measurement parts of the study, and verbal consent will be obtained from focus group interview participants. All methods will be carried out in accordance with the Declaration of Helsinki.

Questionnaire. A custom questionnaire comprised of modules from well-established, validated, and documented questionnaires and modules adapted from questionnaires previously used in our research group to assess other occupational groups (ex. [32, 41] but adjusted to specifically capture information about the warehousing work environment.

The questionnaire modules consider: personal and demographic parameters (6 questions), employment conditions (4 questions), work task competence and work tasks currently performed (2 questions), workload, fatigue and recovery (12 questions–including the short SOFI questionnaire [42, 43]), musculoskeletal symptoms (Standardised Nordic questionnaire [44], 1 question with individual ratings for 10 areas of the body), mental health symptoms (1 question with individual ratings for stress, anxiety and depression), job satisfaction (1 question), and 4 dimensions from the Copenhagen Psychosocial Questionnaire COPSOQ III questionnaire [45], (social support from supervisor, sense of community at work, organisational justice, and work pace; in total 13 questions).

The questionnaire will be provided both online and in paper format, and will be available in Swedish and, if necessary, in English. To the extent possible, participants will be given the opportunity to fill in the survey during working hours.

A unique study code will be issued to all participants and completed questionnaires will only be marked with the study code to ensure anonymity.

**Technical measurements.** Upper arm and trunk elevation angles, physical activity profiles (e.g., sit, stand, walk) and heart rate will be assessed over five consecutive days. Wireless tri-axial accelerometers (Axivity, Newcastle upon Tyne, UK) will be affixed to the skin using double sided toupee tape (between the device and the skin) and breathable medical tape (over top of the device) to assess postures and physical activity. Specifically, upper arm elevation will be recorded using accelerometers mounted on the left and right arms, positioned with the upper edge of each unit aligned with the insertion of the deltoid muscle into the humerus, with the long axes of the unit and the humerus aligned [35, 46, 47]. Trunk elevation angle will be recorded using a unit placed on the sternum with the top edge of the unit at the level of the sternoclavicular notch, and the long axis of the unit aligned with the trunk. A final accelerometer unit will be placed on the anterior surface of the right thigh approximately midway between the hip and knee with the axes of the unit and the femur aligned. The thigh accelerometer will be used to determine body postures (ex. sit/stand/lie) and activities (stand/walk/run/bike) adopted during the data collection period.

To facilitate estimates of relative workload and autonomic regulation (stress level), workers will also wear a BodyGuard heart rate (HR) monitor for recording cardiac activity (ECG) (Firstbeat Bodyguard; Firstbeat Technologies Ltd., Jyväskylä, Finland) [48]. Two electrodes (Ambu, Denmark) will be positioned according to Firstbeat recommendations, and a wireless logger will be attached to the electrodes.

**Diary.** To permit analysis of posture and heart rate for individual tasks, workers will be issued a daily logbook where they will record their working hours including the timing of work tasks and breaks taken each day. A list of work tasks developed a priori in consultation with management will be provided to ensure the tasks are described in the terminology specific to each warehouse.

**Individual interviews and focus group interviews.** To deepen our understanding of working conditions and musculoskeletal discomfort along with any perceived connection to organisational processes and practices, approximately 3–5 focus group interviews will be

conducted with employees at each of the warehouses. An interview guide will be used to drive the semi-structured focus groups and will include themes overlapping with the 4 COPSOQ questionnaire dimensions (namely, social support from supervisor, sense of community at work, organisational justice, and work pace). Participants will be asked to describe and reflect on their work environment (including physical and mental workload and musculoskeletal discomfort, and social relations at the workplace), scheduling of work hours, the assignment (formal and informal) of work tasks, the qualification requirements for performing work tasks, recruitment and career paths and leadership. Participants will also be asked to consider whether they experience (in)equality in relation to those aspects of the work and workplace. Focus group interviews will be approximately 45–60 minutes in duration.

Semi-structured in-depth interviews will be performed with managers, health and safety and/or union representatives at each worksite. The purpose of these interviews is to gain knowledge about each warehouse including their organisational structure and their organisation of work. Questions will be asked to deepen our understanding of recruitment practices, the most challenging work environment issues, and the extent of digital technology use in the warehouse operations system.

Focus groups interviews will be conducted on-site in-person. Individual interviews will be conducted, to the extent possible, in person at the warehouse, and otherwise online. All interviews will be recorded for subsequent transcription.

**Company data.**   Work schedules will be attained for all employees working during the data collection week. In addition, each company will be asked to share their operations systems performance data on loads handled in each task at the most detailed level possible (average or individual worker) for each day during the previous year. For example, for picking workers, this could include individual estimates for: total number of orders picked per day; range in the number of items picked per order; total number of items handled per shift; total weight of items handled per shift. These data will be used to estimate loads at the individual level, as well as to assess seasonal variability in loads.

## Data processing and statistical analysis

**Questionnaire data.**   Questionnaire data will be used to quantitatively map employees' experiences of physical and psychosocial workload and health. Questionnaire responses from the COPSOQ module will be processed in accordance with standard COPSOQ's analysis to create an index for each of the included dimensions (26).

To assess inequalities, questionnaire data will be stratified by sex and, to assess the effect of race/ethnicity, we will stratify based on PoB in two categories: SB vs FB. Mean values for each pair of categories will be compared using traditional statistical processing techniques and regression models to assess the effect of sex and of PoB on perceived physical and psychosocial demands and resources and MSD health. Based on our experience of the spread of these variables between individuals, we estimate that a data material based on approximately 200 employees (estimated response rate of 60–70%) is large enough to achieve sufficient power to be able to demonstrate relevant differences, even in stratification by sex (approximately 50% females) and country of birth (approximately 50% FB), as we have seen in our pilot work in online retail warehouses.

To the extent possible, we will assess differences when stratifying for both sex and PoB by comparing mean values from the four groups (SB men, SB women, FB men, FB women) in perceived physical and psychosocial demands and resources and MSD health.

**Technical measurement data.**   The 5-day accelerometer recordings will be analysed using the open source ActiPass software program (Uppsala University), which uses a combination of ActiFour algorithms [49] together with revised algorithms for identifying periods of lying

down [50]. Periods of non-wear and corrupt data will be identified and eliminated. Remaining data from the thigh-mounted accelerometer will be used to categorize time spent by each worker in activities including, sitting, standing, walking or more strenuous activities and will be expressed in terms of minutes per day, as well as in percentage of working time, and percentage of the total day. Data considering proportions of a whole (ex. Percentages of day) are inherently interdependent, and as such, these data will be processed and analysed using Compositional Data Analysis [51, 52].

Arm- and chest- mounted accelerometer data will be used to determine information about arm and trunk angles and movement speeds, respectively, and data will be reported using the set of comprehensive variables used in previous studies [53–55].

HR data will be downloaded and visually inspected using the Firstbeat SPORTS uploader (V.1.0; Firstbeat Technologies). Periods of corrupt data will be removed, and a continuous data file formed from the set of HR files for each worker. The 5-day heart rate recordings will be analysed using ActiFour [49] algorithms in conjunction with our own analysis software to assess activity in the autonomic nervous system [56].

Task-based analysis. Task-based exposure estimates will be calculated to facilitate consideration of differences between groups stemming from work organisation (cf Fig 1- differences between groups in the tasks performed).

Task-level exposures will be calculated at the level of individual workers for accelerometer-based outcome variables including: mean upper arm and trunk inclination angles, variation in upper arm and trunk angles, and percentage of task time spent sitting, standing and walking; and for HR variables including percent Heart Rate Reserve (%HRR).

Group-level mean exposure estimates with variance components will then be calculated for each task-level exposure variable and work task, as a basis for constructing a task-based exposure matrix at each warehouse. Individual full-day exposures will be determined for all workers at each warehouse by combining individual worker schedules obtained from the company with data from the task-based exposure matrix. With caution, this can then be interpreted in relation to the risk of health problems and to assess differences between groups based on sex and PoB within each warehouse. A sample of 20 employees at each worksite is considered acceptable to build an exposure matrix of adequate reliability, given that the number of tasks per site is small. Analysing the interaction of sex by PoB may not prove possible if the material contains too few people in any of the combinations.

**Individual interviews and focus group interviews.** Following word-by-word transcription of all focus groups and individual interviews, the material will be used to develop a deeper and more nuanced understanding of organisational factors that contribute to patterns of (in) equality identified from the quantitative analyses. The data will be analysed using Acker's theory of inequality regimes which includes both organisational and individual-level factors [39]. This will provide a framework for analysing how complex inequalities are produced and reproduced within each organisation based on practices, processes, actions, and meanings. After an inductive coding of the transcribed interviews, the first-cycle codes will be mapped to Acker's five themes (organising practices and processes, control and compliance, shape and degree of inequality, visibility of inequality, and legitimacy of inequality) as well as to the factors Acker argues provide the most common basis for inequality (class, gender and race/ethnicity). Thus, we will consider whether differences exist based on gender or race/ethnicity in, for example, allocation of work tasks, promotion opportunities, employment security, benefits, salary, control and compliance, among other organisational practices. The analysis will also take into consideration the extent to which inequality patterns vary between different organisations [39].

Throughout the analysis, the aim will be to distinguish patterns of continuity and coherence as well as variations and contradictions in the individual interview and focus group material.

The analysis will seek to understand organisational factors and clarify when and in relation to which specific circumstances, practices and processes gendering and racialization exist in the organisation and in the workers' experiences and understanding of their work and workplace. This means, for example, that the analysis will oscillate between focusing on how work and qualifications are understood and valued, and how, in turn, this is shaped by integrated practices and processes of gendering and racialization, either explicitly or (more often) implicitly.

Mixed methods analysis. Using the framework outlined in Fig 1, we will address our specific study aims using a mixed methods approach. In the quantitative analyses, we explore inequality related to sex by categorizing workers as male or female based on the self-reported data, and we will explore inequality related to race/ethnicity by categorizing workers by country of birth. In the qualitative analyses, we will consider inequality aspects related to gender and race/ethnicity via consideration of organisational practices and processes (re)producing inequality regimes.

To describe working conditions and work-related musculoskeletal health in online retail warehousing (aim 1), data on physical exposures at work will be reported for the sub-population participating in technical measurements including task-level and full work-day measurements of: physical activity compositions, arm and trunk angles and movement speeds, and autonomic nervous system activity. From the group of workers who complete the questionnaire, data on psychosocial exposures and musculoskeletal health will be included. Finally, task-based estimates of loads handled determined from company data will be included.

To determine the extent to which working conditions and work-related musculoskeletal health differ between employees with respect to sex and race/ethnicity (aim 2), questionnaires, technical measurements and company data will be used. For the set of employees for whom we have sex and PoB data (from the questionnaire), full day exposure profiles estimated from the company scheduling data in conjunction with the task exposure matrix will be used to assess differences in physical and psychosocial exposures and MSD health based on sex and race/ethnicity.

To describe the extent to which eventual sex/gender and race/ethnicity inequalities in working conditions and musculoskeletal health are related to organisational processes and practices (aim 3), technical measurements, company data, questionnaires and focus group and individual interviews will be considered together. We will use data from the qualitative analyses to deepen the understanding of what drives differences in the quantitative survey and physical exposure data including consideration the organisational processes and practices that may underlie eventual differences in working conditions and MSD health based on sex/gender and race/ethnicity.

### Advisory committee

Throughout the study, researchers will consult an advisory committee comprising representatives from the Swedish Trade Federation, the Union of Commercial Employees, the Swedish Work Environment Authority, and several warehouses. The advisory committee will be used for discussions regarding practical items relevant to the implementation of the study, for example, establishing a contact network for the recruitment of warehouses. Further, the advisory committee can contribute to the interpretation of results in a practical context and in disseminating results within the online retail sector upon completion of the study.

### Discussion

This study employs a novel approach that combines established techniques from gender research for assessing organisational perspectives on sex, gender, race, ethnicity and (im)

migration status, with a quantitative analysis of the consequences of said perspectives. Together, this mixed methods approach permits consideration of gendered and racialized processes that can lead to differences at the level of the individual worker in terms of employment conditions, facility design (e.g., differences in experiences with digital technologies), work organisation (e.g. task allotment), work environment conditions (physical and psychosocial), and MSD health, as outlined in Fig 1.

Warehousing comprises a range of work tasks which differ in terms of complexity and workload. Thus, it is important to consider potentially inequalities in task distribution based on sex/gender and race/ethnicity and subsequent health effects. In this study, we will include warehouses employing 20–200 production employees. We believe this is a sufficiently size to provide access to an adequate number of male and female and Swedish- and foreign- born workers to permit assessment of inequality regimes based on both sex/gender and race/ethnicity. Further, we hypothesize there will be sufficient inequalities in task distribution and thus job exposure to have the potential to impact occupational musculoskeletal health.

We will aim for a response rate of 60–70% for the questionnaires, which is also in line with previous field experiments. To maximize our chances of reaching this response rate, we will offer the questionnaire in paper and digital format (computer and mobile phone compatible) and will strongly encourage warehouses to permit employees to respond during working hours. Based on prior field research, we believe it is realistic to recruit 20 workers at each warehouse to participate in the technical measurements of physical workload, and 20 workers to participate in focus group interviews. As stated above, these predicted sample sizes are expected to allow stable estimates to the research questions.

The main limitation of the study is a limited generalizability of the findings given that we only investigate three warehouses while the range in types and sizes of warehouses is large. However, we believe that findings from our case studies will be useful to other similarly sized warehouses with diverse employee populations and will inspire warehouses to consider their own organisational practices in terms of inequalities in sex/gender and race/ethnicity.

## Dissemination

The results of the project will be shared directly with the employees and managers at the participating warehouses, the union representing the workers and the advisory committee via presentations and summary reports. Research results will also be disseminated via publication in international, peer-reviewed, open-access scientific journals and presentation at international scientific conferences.

## Author Contributions

**Conceptualization:** Jennie A. Jackson, Svend Erik Mathiassen, Klara Rydström, Kristina Johansson.

**Data curation:** Jennie A. Jackson, Klara Rydström, Kristina Johansson.

**Formal analysis:** Jennie A. Jackson, Klara Rydström.

**Funding acquisition:** Svend Erik Mathiassen, Kristina Johansson.

**Investigation:** Jennie A. Jackson, Klara Rydström.

**Methodology:** Jennie A. Jackson, Svend Erik Mathiassen, Klara Rydström, Kristina Johansson.

**Project administration:** Jennie A. Jackson, Klara Rydström, Kristina Johansson.

**Supervision:** Jennie A. Jackson, Svend Erik Mathiassen, Kristina Johansson.

**Writing – original draft:** Jennie A. Jackson.

**Writing – review & editing:** Jennie A. Jackson, Svend Erik Mathiassen, Klara Rydström, Kristina Johansson.

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
