## [Decision Letter · Decision Letter 0]

7 Nov 2023

PONE-D-23-06460Protocol for an observational study of working conditions and musculoskeletal health in Swedish online retail warehousing from the perspective of gender and race/ethnicity.PLOS ONE

Dear Dr. Jackson,

Thank you for submitting your manuscript to PLOS ONE. After careful consideration, we feel that it has merit but does not fully meet PLOS ONE’s publication criteria as it currently stands. Therefore, we invite you to submit a revised version of the manuscript that addresses the points raised during the review process.

We look forward to receiving your revised manuscript.

Kind regards,

Amin Yazdani

Academic Editor

PLOS ONE

Journal Requirements:

“This work was supported by the Swedish Research Council for Health, Working Life and Welfare (Forte) grant number 2019-01051.  “

“KJ & SEM - Swedish Research Council for Health, Working Life and Welfare (Forte) grant number 2019-01051.  https://forte.se/

The funders did not and will not have a role in study design, data collection and analysis, decision to publish, or preparation of the manuscript.”

Reviewers' comments:

Reviewer's Responses to Questions

**Comments to the Author**

1. Does the manuscript provide a valid rationale for the proposed study, with clearly identified and justified research questions?

Reviewer #1: No

2. Is the protocol technically sound and planned in a manner that will lead to a meaningful outcome and allow testing the stated hypotheses?

Reviewer #1: No

3. Is the methodology feasible and described in sufficient detail to allow the work to be replicable?

Reviewer #1: No

4. Have the authors described where all data underlying the findings will be made available when the study is complete?

Reviewer #1: No

5. Is the manuscript presented in an intelligible fashion and written in standard English?

Reviewer #1: Yes

6. Review Comments to the Author

You may also provide optional suggestions and comments to authors that they might find helpful in planning their study.

Reviewer #1: Overall comments

This protocol describes a timely project that fills a real gap in working life knowledge in Sweden: given the recent rise in both online retail and immigration, what are the emerging work environment and health issues in this sector? While gender inequities have been broadly studied in Sweden, there is a particular gap in ethno-racialized and intersectional inequalities. The results will be important contribution to a growing issue in Sweden and elsewhere.

I will note that my review is partial and incomplete. It appears that the in-text citations are numbered, and the bibliography is alphabetized by author. Since I cannot decipher what is being referenced, I cannot provide a full review on the suitability of referenced texts or integration of past literature.

Below are my comments on the content of the manuscript, irrespective of references

Comments on frameworks and concept definitions

The authors state throughout the text that comparisons are being made and inequalities discovered relative to ‘race/ethnicity’. The definitions of race and ethnicity come on page 8, and it seems that the socially-constructed definition used here is more in line with the term ‘ethnoracialization’ or ‘racialialization’. I would suggest using those later terms instead of ‘race/ethnicity’ since it will more readily understood over those first 8 pages that the authors are not using an older definition of race which pertains to innate biological traits. This also seems like a better match with ‘gender’, which is widely understood to be a social construct and the two terms are often used together in the text.

But moreover… ethnoracialization is not under study in this paper, except through the surrogate measure of country of birth. Given that there are many ethnoracialized persons born in Sweden, country of birth will not capture the differential experience of those with, for example, darker skin or minority religion that fit in the ‘SB’ category. Although the questionnaire methods state that there are 6 demographic questions, it is not stated whether any of them address ethnoracialization (though it seems unlikely given Swedish restrictions on collecting this type of data). If ethnoracialization is not assessed in any way other than country of birth, then the contribution of the study is relative to country of birth and it should be stated clearly as such throughout (currently the first clear mention that only a surrogate measure is used appears on page 18). In the discussion or methods it could be stated that country of birth can be interpreted as a surrogate measure for ethnoracialization and/or language skills; this interpretation could be augmented/triangulated with qualitative data.

Related to this, I wondered in the introduction if the cited physical size differences of 22cm are relevant to the birth countries likely to be represented among the participant group – but physical size or other physical characteristics related to country of birth do not seem to be captured in the study, and in any case would seem to undercut the later statement that it is the socially constructed categories that are of interest.

Figure 1 is a little unclear, specifically what ‘linkages’ the orange arrows from the orange ‘inequality box’ are meant to describe. The shape is reminiscent of illustrations of confounding, but descriptions of stratified analyses in the text seem to suggest that this is more about differential relationships between risk factors on the left and health outcomes on the right. Perhaps adjustments to the figure or clarifications in the figure caption would help?

The text describes work exposure differences related to both gender (e.g. task allocation) and sex (e.g. height and body size characteristics which are not well accommodated by work stations and tools). However, figure 1 mentions only gender. Which is correct?

Methods

I encourage the authors to consider the term ‘participants’ rather than ‘subjects’. Research teams typically reside within hegemonic categories of class, citizenship, language and ethnoracialization, so reducing the agency of study participants to ‘subjects’ seems unjust and counter-productive (and in any case not in keeping with the respectful tone the paper has elsewhere).

P 13 consider adding ‘purposively selected’ to the description of participant selection to more clearly indicate that this is a non-probabilistic approach sample: “interviews, participants will be purposively selected from the pool of interested parties working day”

P13 has a description of sampling to balance between ‘male’ and ‘female’. These categories are related to sex, not gender; gender equivalent terms are ‘man’ and ‘woman’. This brings up a further point: is there a plan to explicitly exclude gender-diverse categories? If not, consider stating ‘gender-balanced’ which allows for inclusion of trans- and/or non-binary persons as part of the gender story.

What is the anticipated duration of the focus groups and interviews?

Note that a ‘balanced’ group in focus groups may reduce candor by reproducing existing hegemonies. What will be the approach to facilitate candid feedback from all group members?

Interview data analysis

The authors have provided a theoretical framework for the analysis of qualitative focus group and interview data, but have not referred to any specific qualitative approaches or methodologies. It seems that the approach is deductive tested for fit with Acker’s framework with vague allusions to iteration, but it is not clear how the text would be coded, categorized, condensed, what level of description/interpretation is anticipated. It would be helpful to the reader to state clearly the methods/methodological approaches and not just the theoretical standpoint.

Mixed methods

Quantitative comparisons and described between domestic and foreign-born participants – is it also the intention to compare quantitatively by gender?

Minor points

P7: “Even among workers perform the same tasks, the loads” should read:

“Even among workers performing the same tasks, the loads”

P8: In the statement: “Further, differential experiences of psychosocial loads between FB and domestic-born workers including lower workplace decision latitude (7,35)…” it would be helpful to the reader to state the context of the study, since the FB category can mean different things in North or South America, or Europe, or the Middle East.

P14 Nordic Musculoskeletal Questionnaire is broader than just pain; consider listing ‘musculoskeletal symptoms’

P16 “A list of work tasks developed a priori in consult with” should read:

“A list of work tasks developed a priori in consultation with”

7. PLOS authors have the option to publish the peer review history of their article (what does this mean?). If published, this will include your full peer review and any attached files.

Reviewer #1: No

---

## [Author Response · Author response to Decision Letter 0]

4 Dec 2023

2023-11-24

Dear PLOS One Team 

Thank you for the chance to revise our manuscript, PONE-D-23-06460, entitled, Protocol for an observational study of working conditions and musculoskeletal health in Swedish online retail warehousing from the perspective of gender and race/ethnicity.

Please find a summary of our responses to all comments attached as a *response to reviewers* document which accompanies this resubmission. A marked copy of our revised manuscript with the changes highlighted is also included in our resubmission along with a clean copy of our revised manuscript. 

The format of the submitted article meets PLOS ONE's style requirements. All funding-related text has been removed from the manuscript. We request that the following revision to the Funding Statement be made: 

Current text: “KJ & SEM - Swedish Research Council for Health, Working Life and Welfare (Forte) grant number 2019-01051. https://forte.se/

Desired text: “KJ & SEM - Swedish Research Council for Health, Working Life and Welfare (Forte) grant number 2019-01051

Best regards, 

Jennie Jackson

---

## [Decision Letter · Decision Letter 1]

9 Jan 2024

Protocol for an observational study of working conditions and musculoskeletal health in Swedish online retail warehousing from the perspective of sex/gender and place of birth.

PONE-D-23-06460R1

Dear Dr. Jackson, 

We’re pleased to inform you that your manuscript has been judged scientifically suitable for publication and will be formally accepted for publication once it meets all outstanding technical requirements.

Kind regards,

Amin Yazdani

Academic Editor

PLOS ONE

Additional Editor Comments (optional):

Reviewers' comments:

Reviewer's Responses to Questions

**Comments to the Author**

1. Does the manuscript provide a valid rationale for the proposed study, with clearly identified and justified research questions?

Reviewer #1: Yes

2. Is the protocol technically sound and planned in a manner that will lead to a meaningful outcome and allow testing the stated hypotheses?

Reviewer #1: Yes

3. Is the methodology feasible and described in sufficient detail to allow the work to be replicable?

Reviewer #1: Yes

4. Have the authors described where all data underlying the findings will be made available when the study is complete?

Reviewer #1: Yes

5. Is the manuscript presented in an intelligible fashion and written in standard English?

Reviewer #1: Yes

6. Review Comments to the Author

You may also provide optional suggestions and comments to authors that they might find helpful in planning their study.

Reviewer #1: The authors have done a thorough job responding to the reviewer comments, and the updates to the paper improve the clarity. I have no further comments and will look forward to reading the eventual findings related to this protocol.

7. PLOS authors have the option to publish the peer review history of their article (what does this mean?). If published, this will include your full peer review and any attached files.

Reviewer #1: No

---

## [Editor Report · Acceptance letter]

13 Feb 2024

PONE-D-23-06460R1 

PLOS ONE

Dear Dr. Jackson, 

I'm pleased to inform you that your manuscript has been deemed suitable for publication in PLOS ONE. Congratulations! Your manuscript is now being handed over to our production team.

Kind regards, 

on behalf of

Dr. Amin Yazdani 

Academic Editor

PLOS ONE